# Posting Recommendations in Healthcare Q&A Forums

**Yi-Ling Lin [1]** , **Shih-Yi Chien [1],\* and Yi-Ju Chen [2]**

1   Department of Management Information Systems, National Chengchi University, Taipei 11605, Taiwan; yl_lin@nccu.edu.tw

2   Department of Computer Science, The George Washington University, Washington, DC 20052, USA; sabrina11068@gmail.com

\*   Correspondence: sychien@nccu.edu.tw

**Abstract:** Online Q&A forums, unlike search engines, allow posting of various types of queries, thus attracting users to seek information and solve problems in specific domains. However, as insufficient knowledge leads to incomprehensible queries, unsuitable responses are common. We develop posting recommendation systems (RSs) to support users in composing reasonable posts and receiving effective answers. The posting RSs were evaluated by a user study containing 27 participants and three tasks to examine if users engaged more in the question generation process. Two medical experts were recruited to verify whether professionals can understand and answer posts supported by RSs. The results show that the proposed mechanism enables askers to produce posts with better understandability, which leads experts to devote more attention to answer their questions.

**Keywords:** question-answering forum; healthcare informatics; recommendation system; word embedding; user study

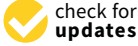

## 1. Introduction

Although search engines are the most popular channel for information retrieval, the retrieved results are often too general to find solutions that fulfill user needs. Information retrieved from search engines is usually selected and sorted using custom algorithms, which favor preselected hosts or Wikipedia results. When looking for information on an unfamiliar topic, users may lack the knowledge to formulate good search queries, resulting in improper or unexpected search results. The difficulty in composing concise queries for search engines has popularized online Q&A forums, which serve as alternatives by which to find detailed answers to questions. Online Q&A websites attract users because they can respond to detailed questions and query experts without time or geographical constraints [1]; however, for user questions that are incomplete or ambiguous, the resulting answers may not be what the user was looking for; finding professional and reliable answers can be difficult. This has led to many unsolved and unclear questions in online forums.

Generating effective questions on Q&A websites is not easy, particularly for highly specialized domains. In the healthcare field, for instance, people may possess little background on the questions and may not understand the relevant jargon, resulting in ambiguous questions. Most users can only think of simple terms to describe their disease and medical conditions: Phrases used in the queries often do not reflect standard medical terminology. Sometimes, even the asker is not sure how to describe his/her medical condition or to describe the encountered situation in various ways (e.g., different descriptions of the pain scale for the same illness) [2]. Lexical barriers such as partial misspellings and the use of abbreviations also makes questions hard to understand. For example, a typical general question is "Recently I have been suffering from back pain. What kind of lifestyle would help prevent back pain?" A more informative or knowledgeable post would be "I am staying at a healthy weight, but I recently began to suffer from severe back pain. I searched

for information online and found that smoking ages the spine. I seldom smoke but my husband smokes a lot. Is it because of inhaling too much secondhand smoke?"

Since this difficulty in formulating effective posts on Q&A websites is rarely addressed, we propose a design that recommends more concepts (e.g., topics and terminology) to users to formulate their posts with more reasonable details, rather than presenting existing questions from the search pool (i.e., routing answers) or finding possible answerers as do most Q&A websites [3–5]. While most products provide recommendations to users at querying and browsing moments, only a few mechanisms (e.g., spellchecks) focus on helping users formulate posts, particularly for an online Q&A forum with a specific subject such as healthcare. Referring to the research thread [6–8], enhancing the quality of the input content not only increases the user's ability to get useful answers but also results in high-quality solutions faster. Recommendation systems (RSs) applied when composing posts could be enhanced by suggesting to users what content should be posted and how to describe the situations in the post, leading to a high input quality and better answers [8]. In this study, we seek to help participants who are unfamiliar with a domain to compose queries with a posting recommendation mechanism.

We propose two posting RSs: a word embedding-based and a semantic-based RS. Word embedding is a well-known tool for processing words into space vectors to improve the automatic understanding of human languages. Our word embedding-based posting RS (we use "the embedding model" in the following content), implemented by a Word2Vec model [9], is trained on 5319 questions and 500 abstracts of publication crawled from health-related websites. For the semantic-based posting RS, we adopt the WordNet (https://wordnet.princeton.edu/) model (we use "the semantic model" in the following content), a lexical database for English with several synonyms that are tagged artificially. It groups words from their meanings for computational linguistic and natural language processing (NLP). Both the embedding and semantic models are meant to recommend ideas and terminology that users may need in their current posts. These feature-based recommendations are expected to help users make more subject-specific posts.

We believe that using text analytics to participate more in the asking processes can be a good approach to support users formulating posts and enhance the clarity of the posts, which would encourage domain experts to reply to the posts and answer the questions. To verify whether reformulated queries yield better query wording and help users to find the desired answers more easily, we conducted a user study and a satisfaction questionnaire to understand user perspectives on our RSs. In addition, posts written by our study's participants were evaluated by experts with a health-related background to determine whether they could be easily solved. The research questions are posited as follows:

RQ1: Does the posting RS help users formulate questions in healthcare Q&A forums?
RQ2: Is it easier for experts to understand questions supported by the posting RS?

## 2. Related Work

Traditionally, healthcare professionals are the primary sources of health information; they provide and manage health information for their patients [10]. With the spread of the Internet, sources of health information have become more diverse and accessible to individuals and families. Despite this easy access to health information, its main use remains focused on supporting healthcare professionals, such as in hospital information systems [11]. Isern and Moreno [12] organize various agents in healthcare to inform decisions on cure plans and to alert patients when abnormal messages are detected. Although health information has been widely applied to support professionals, Frost et al. [13] state that health information is also beneficial for patients and people in need. Effective support in terms of health information can improve the doctor–patient relationship as well as the completeness and quality of diagnosis [13]. Thus, it is crucial to provide an effective communication channel between professionals and general users in the health information domain.

As the Internet provides a convenient way to access health information, people tend to seek health-related support online [10]. It is estimated that approximately 12.5 million of the 278 million daily Internet searches are health-related [14]. To find the most relevant answers on the Internet, RSs are essential for Q&A forums. Existing RSs generally focus on routing answers or finding answerers. Among various recommendation mechanisms, question routing and grouping are two main approaches to finding potential answers and answerers (people who have similar experiences in a specific area) in Q&A forums [3–5]. These methods consider underlying social network features (e.g., which query gets more hits), user activity (e.g., which category do experts tend to be active in and receive honor for the best answer), and public personal data on websites to improve system usability.

Most studies about RSs in online Q&A forums focus on general aspects rather than a specific subject such as healthcare. Budalakoti et al. [15] present a RS with three different methods for selecting the most appropriate responder given a question on Yahoo! Answer. One is calculating the cosine similarity between the words from an individual's (the author) historical Q&A data and his/her current question; another is grouping documents using K-means clustering; and the other is discovering the author-topic distributions as the general model and recommending the responders based on the marginalized probabilities. Yang and Amatriain [16] analyze the application of RSs at Quora and build a platform to experiment with different machine learning models for the developers. While most studies work on general Q&A forums, few studies focus on specific professional Q&A forums. Xin et al. [17] developed TagCombine, an automatic tag recommendation method to analyze objects in both Stack Overflow and Freecode websites to facilitate search and identify software objects. Pedro and Karatzoglou [18] presented a supervised Bayesian approach to model expertise with similar topics to support question recommendation and to avoid question starvation from the Stack Exchange (http://stats.stackexchange.com). Wang et al. [19] also provided an enhanced tag recommendation system, ENTAGREC$^{++}$, for organizing questions and facilitating browsing questions on Stack Overflow. Singh and Simperl [20] implemented a system, Suman, which combines semantic keyword search with traditional text search to find answers for unanswered questions on Reddit and Stack Overflow. There are even fewer studies focus on healthcare Q&A domain. McCray et al. [21] developed a web-based terminology server which allows a diverse audience to easily access current health information by enforcing flexible query grammar, expanding synonyms and lexical variants for a term, and generating alternative spellings for unknown words. Cho et al. [22] helped users to receive satisfactory responses by improving the baseline retrieval model with semantic information to generate top 5 discussion threads that are potential responses for unresolved medical case-based queries. Although RSs have been widely employed in health areas, Jacobs et al. [10] state that the extant mechanisms for online health information search are insufficient.

Despite the popularity of Q&A forums, many questions lack answers due to ambiguous or misleading terms [20]. Baltadzhieva and Chrupała's study [8] on Stack Overflow (a programming Q&A forum) shows that the terms used, tags added, and the length of questions influence question quality. They conclude that questions that are too localized or that have incorrect tags or terms are considered to be of poor quality [8]. In the healthcare domain, Bochet et al. [23] demonstrated that most users are too inexperienced to formulate an effective search query on health information. Spink et al. [24] also showed that when posting medical and health queries, many users fail to retrieve information relevant to their condition due to an ignorance of specialized vocabulary or precise medical terms. Zhang [25] showed that queries posted about health support are usually simple and short and lack other aspects of individual information. For recommendation systems to facilitate the formulation of online questions that are more likely to be answered, it is essential to make posts more comprehensive.

Thus, in this study we focus on generating and improving questions to enhance the recommendation mechanisms in the healthcare domain. We develop posting RSs to suggest potential ideas, formulate user questions, and eliminate ambiguities that might decrease

the likelihood of the question being answered or increase the time it takes for the question to be responded to.

## 3. Posting Recommender Systems (RSs)

### 3.1. Interface Design

After looking over Q&A online forums (e.g., Quora.com, Yahoo! Answer, Stack Overflow (https://stackoverflow.com/), and English Language & Usage (https://english.stackexchange.com/)), we included an input area and a recommendation area in the system layout (see Figure 1). The first column of the recommendation area (the table part) shows topics that askers may focus on and the rest of the columns show the top 10 terms related to the particular topic.

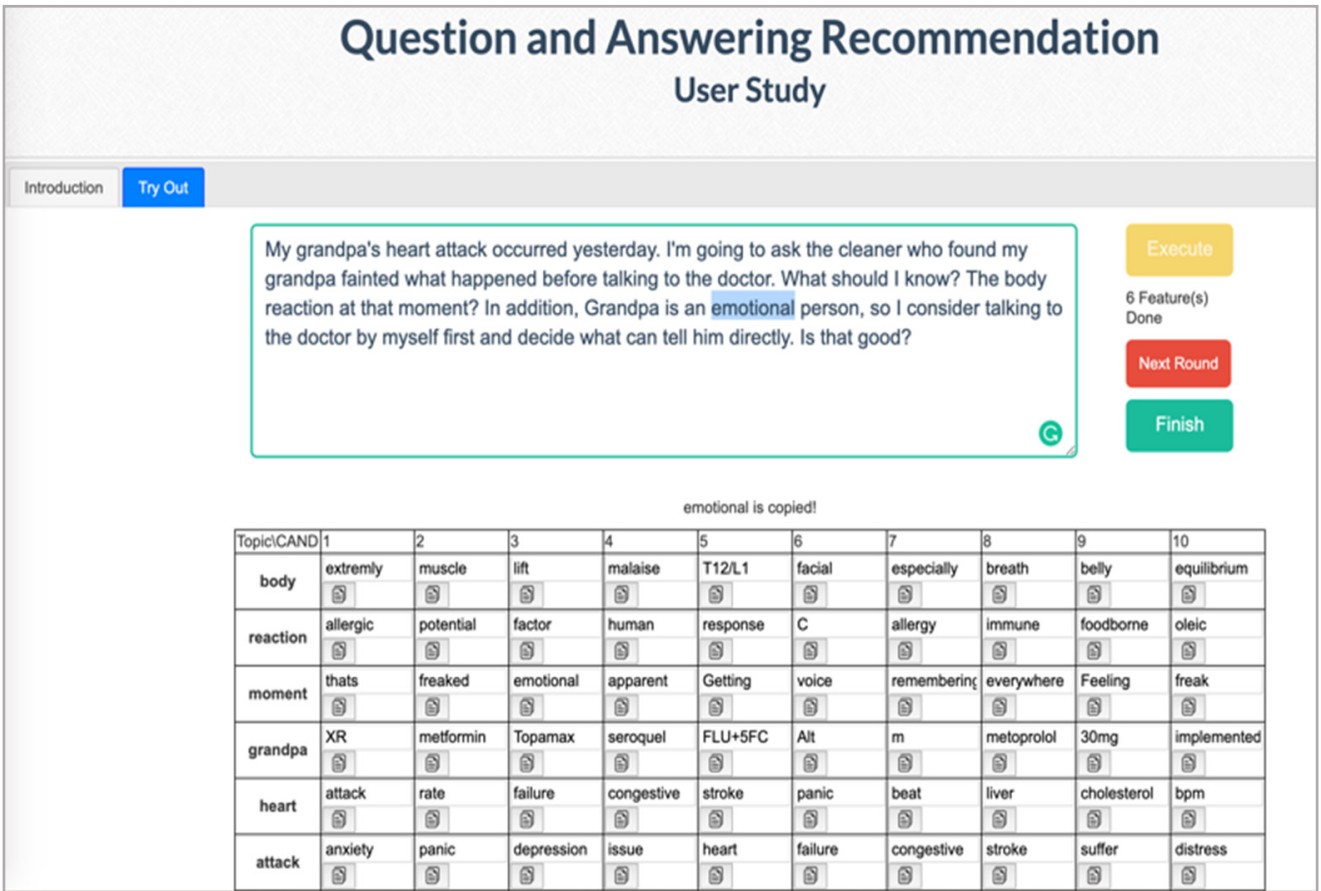

**Figure 1.** Interface of post RS.

Askers compose multi-sentence posts in the green input region (Figure 1). While users compose their posts, Grammarly (https://www.grammarly.com/), an auto-spellcheck extension from the Chrome web store, is activated to eliminate careless typos. If askers need ideas or assistance in generating the appropriate terms to pose their questions, they click on execute to receive system suggestions. In the recommendation table, askers click on the copy button to fetch the required terminology. The askers can click on execute at any time to receive new system recommendations. When askers are satisfied with the post, they click on finish to accomplish the question content. Figure 2 demonstrates how users interact with the proposed RSs.

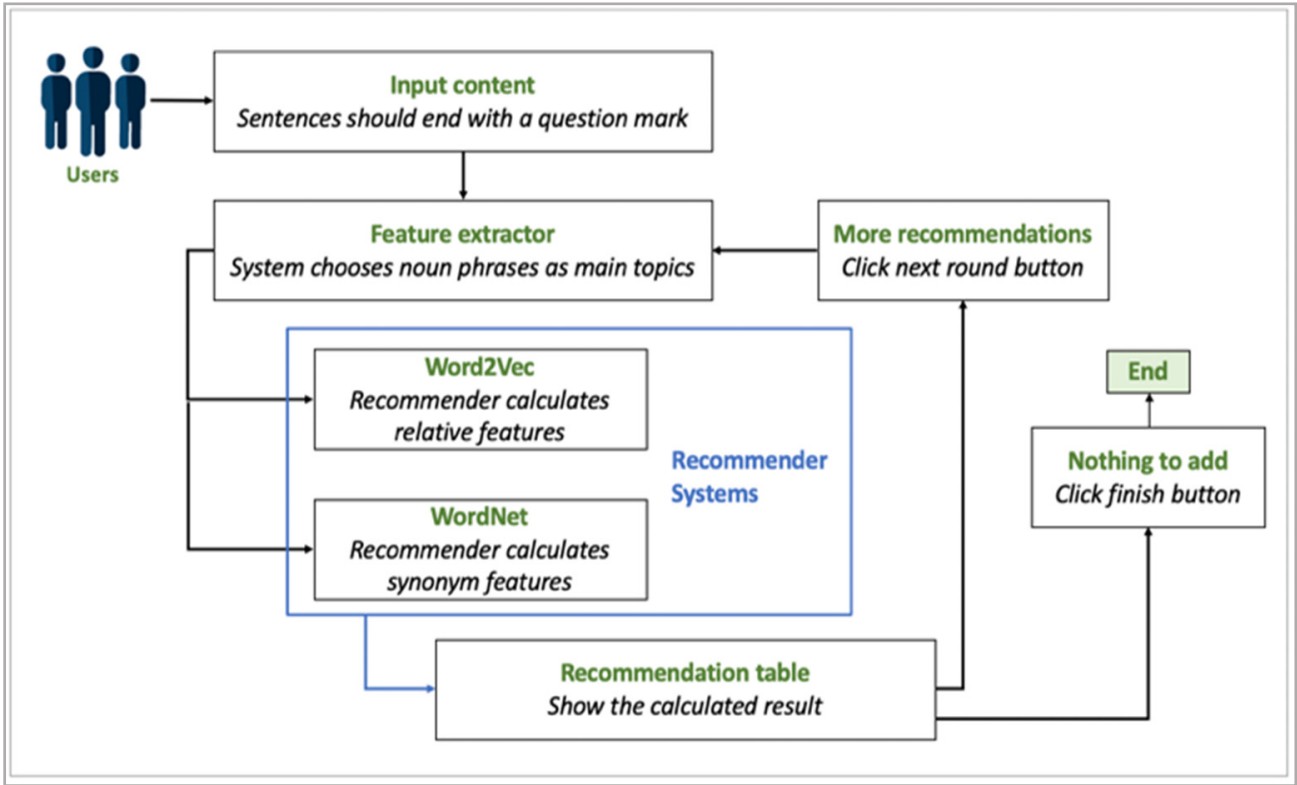

**Figure 2.** User perspective of interactive data flow.

The post recommendation mechanism is composed of three phases. First, the user inputs keywords, terms, or sentences to describe her questions. After receiving the user's queries, we attempt to understand what the user is asking or what concepts she is interested in. A post on Q&A forums is a kind of user-generated content (UGC) usually consisting of a question or a narrative. To identify user intentions from posts, we use a noun phrase extractor to extract the main topics from each post. Noun phrases are usually the core topics or objects in a sentence, whereas verb phrases describe actions between the objects in a sentence.

Second, we use embedding and semantic models to provide recommendations to help users construct their posts. In this study we use Word2Vec, a two-layer neural network model [9], as the embedding model. In addition, as the semantic model we use WordNet (https://wordnet.princeton.edu/), a well-processed English lexical database. We implemented the application using Python NLTK's WordNet package to generate recommendations. Both embedding-based and semantic-based recommendations are triggered by clicking on execute (more details about the recommendation models and dataset are provided in Sections 3.2 and 4.1.)

Thirdly, the recommendation made by these models is prioritized and displayed with the top 10 recommended terminology, where the users can fetch the required content. They continue to modify the post (such as sentences and terms) until they are satisfied with the post, or until nothing new comes out from the recommendations.

### 3.2. Recommendation Models

Several state-of-art recommendation methods such as content-based [26–28], collaborative filtering-based [29] and hybrid methods [30] have been proposed to generate personalized recommendations based on the relationship between users and items. To provide recommendations to help users to construct their posts, we use embedding-based and semantic-based RSs that concentrate on interactive items (i.e., posts) without knowing the previous interactions of the user. To algorithmically understand the post and provide

recommendations, text representation is important. Different from the traditional text representation such as continuous bag-of-words or Term Frequency-Inversed Document Frequency (TF-IDF) [31], WordNet and Word2Vec bring extra semantic features that help in identifying textual content. WordNet [32,33] is a human-curated ontological symbolic representation based on the similarity between words. It is often limited with its hierarchical representation. Word2Vec [9] is an unsupervised neural network method for determining words meaning by its surrounding context with a vector. The input words are transferred into an n-dimensional vector space, then similar words are identified by being near the input vector. It can perform effectively no matter how many words are included in the input vector, but is constrained by the corpus to the vector space. By incorporating the generalizable contexts into the model, Word2Vec has been proven to be more accurate than other models [9,34,35].

Pre-processing is essential for RS models. Data collected from websites and online forums often contain colloquial sayings and abbreviations (e.g., please → plz, pls). To eliminate meaningless words and punctuation (e.g., "?", ".", ";"), we tokenized sentences, removed stopwords, and regulated terms from the NLTK corpus before training. To reduce the number of inflectional forms, we lemmatized the words (e.g., am, are, is → be) using NLTK to get the general patterns of words. We then put all of the word packs of each sentence into a collection and used the gensim package (https://pypi.org/project/gensim/), a Python Library for scalable statistic semantics.

To give suitable ideas to help users compose their posts, we developed two models. We implemented the embedding model using Word2Vec, a shallow, two-layer neural network model that uses a large corpus of texts to perform unsupervised learning [36] and produces a vector space to reconstruct the linguistic contexts of words. In the new vector space, words sharing common contexts in the corpus are located in close proximity to each other. Vector relationships can be represented as "Kitten:Cat = Puppy:Dog". Thus, given expressions such as "Kitten:Cat = Dog: ?", we can infer what words should be inserted. In addition, there are two kinds of Word2Vec models: skip-gram (infers context words based on input words) and continuous bag of words (CBOW: infers input words based on context words). In this work, we followed the gensim tutorial (https://radimrehurek.com/gensim/tutorial.html) and used skip-grams to train Word2Vec on a corpus of medical terms and healthcare forum wording.

We also implemented another recommendation system using the WordNet semantic model. We did not change this much because its database is already well-organized. The recommendations are generated based on the English lexical database using the Python NLTK WordNet package given the input sentence.

We utilized selenium-web browser automatio (https://www.seleniumhq.org/) to support users to eliminate misspellings when formulating their posts. When a user types a period or clicks the execution button, the system considers the prior section to be a sentence, automatically normalizes their wordings and feeds them into two recommendation models. The embedding model would map the input words to its context word and offer recommendations. The semantic model would map the input words to the semantic graph of lexical items it pre-generated and then provide similar terms as recommended ideas.

## 4. Research Design

To assess whether the proposed posting RSs help users formulate queries that increase the probability of being answered, we conducted a user study to collect and analyze content written by users. We implemented two posting RSs: a word embedding model based on Word2Vec (suggesting ideas (terminology) related to the main topics of the input content), and a semantic WordNet-based model (suggesting synonyms (terminology) for the main topics of the input content), for comparison with the baseline model (no recommendation). We collected the participant behavior and posts using the three models for further analysis and expert evaluation.

### 4.1. Dataset

WebMD (https://www.webmd.com/) is one of the few healthcare Q&A forums in which medical specialists (called experts in the forum) offer suggestions to askers about their illness or concerns. The dataset was crawled from WebMD from March 2010 to September 2014 and contains 25,319 questions.

Apart from the daily conversations from WebMD, we also collected medical terminology and specialist wording from other professional healthcare-related websites (such as PubMed (https://www.ncbi.nlm.nih.gov/pubmed/)). Lai et al. (2016) suggest that for word embedding models, the domain of the corpus is more important than its size. Thus, we crawled the abstracts of biotechnology-related publications from PubMed to create the Word2Vec model.

### 4.2. Tasks and Experimental Materials

To generate posting ideas for the participants, the experiment provides a short introduction with a background story to simulate possible healthcare conditions. To complete the task, the participants were asked to compose a post associated with the background story.

To evaluate the experimental design, a pilot test was conducted in which three health tasks were examined: flu, asthma (https://www.webmd.com/a-to-z-guides/common-topics), and pregnancy. The results showed that it was difficult for participants to compose posts about asthma and pregnancy because they had little daily experience in these areas. Therefore, we changed the selection of health tasks to flu, allergy, and foodborne illness, which are more common among the public, and employed these in the official study (see Appendix A).

As a short introduction lacks sufficient information to formulate posts under a simulation condition, for each task we prepared supportive paragraphs from relevant medical websites. To cover various aspects of health situations, articles, news, and reports from healthcare agencies were collected as our materials. Finally, we selected supportive excerpts from a health agency's announcement with statistical data (the rate of an illness in a region) and sections gathered from news reports with common knowledge that the public can understand. Participants were to imagine the assigned task and write down their own or the character's experiences of specific illness after understanding the background information.

In addition, we prepared an example question in the try-out (Figure 3) to encourage participants to produce longer questions and not simply question sentences like "what are the symptoms of heart disease."

### 4.3. Participants and Procedure

Twenty-seven participants (14 females and 13 males, average age 27.7) were recruited from a social media website (i.e., Facebook). Fifteen out of the 27 participants' experience with Q&A forums was limited to browsing discussion threads, rather than composing or answering posts. Only five participants had experience using "professional" Q&A forums.

The within-subject design was used in the experiments. The three tasks were performed along with three algorithmic models (without RS, with Word2Vec RS, and with WordNet RS). Thus, each participant was asked to complete a total of six posts in three tasks. The Latin square design was applied to avoid the order effect [37]. The experiment used the following procedure (Figure 4):

(1) After signing the consent form, the participants took the pre-test questionnaire on their background and past experience using Q&A forums.

(2) A training task was then provided to ensure that participants fully understood the experimental systems and task requirements. An example of an expected post was given to encourage the participant to compose complete questions. Participants were allowed to ask any questions during this step.

(3) A brief description of the assigned model was also provided. The participant was given sufficient time to become familiar with the system.

(4) A description of the general context of the assigned task was provided to the participant, after which the participant began her posting.

(5) Another description of the complex context of the task was given to the participant. Then, the participant began her posting. Please note that as each participant completed all three tasks with the three models, she completed (3)–(5) three times.

(6) A post-questionnaire was issued to the participant to evaluate each user's experience with each model, including her perception of the system process, system speed, and the extent to which they would prefer using our RSs.

---

*[Task 1]*

**Background:**

Sandy's Grandfather has a family history of the heart attack. Unluckily, his illness occurred yesterday and was sent to the hospital. After receiving a phone call from Dad, Sandy tried to search for some information about the sickness. She will go to pick up Grandpa Johnson tomorrow on her way home but she has no ideas what she should know in advance. The following is the information she has now. If you are Sandy and want to get help on the health care online forum, what you will say?

**Supportive paragraphs of a daily scenario task:**

A guide to a heart attack

When blood can't get to your heart, your heart muscle doesn't get the oxygen it needs. Without oxygen, its cells can be damaged or die. Over time, cholesterol and a fatty material called plaque can build up on the walls inside blood vessels that take blood to your heart, called arteries. This makes it harder for blood to flow freely. Most heart attacks happen when a piece of this plaque breaks off. A blood clot forms around the broken-off plaque, and it blocks the artery.

The following is the call, from Sandy's dad:

"If Tracy (paid cleaner) wasn't there at that time, it may have been too late to rescue your grandpa. You know, Grandpa Johnson had a heart attack. He told me before that his chest was sometimes painful and that made it difficult for him to breath. And our hometown was pretty cold in the winter. I'm afraid that if Grandpa forgets to dress warm enough, the low temperature may stimulate another heart attack. Do you think I should find a personal physician for grandpa? Near his house? We are all working outside the county. When emergency happens, this protection may work."

**Example question from Sandy:**

My grandpa's heart attack occurred yesterday. I'm going to ask the cleaner who found my grandpa fainted what happened before talking to the doctor. What should I know? The body reaction at that moment? In addition, Grandpa is an emotional person, so I consider talking to the doctor by myself first and decide what can tell him directly. Is that good? By the way, the temperature here is pretty low. Does anyone know what things should be prepared for when grandpa goes back home?

---

**Figure 3.** Material read by participants before composing a post in the try-out.

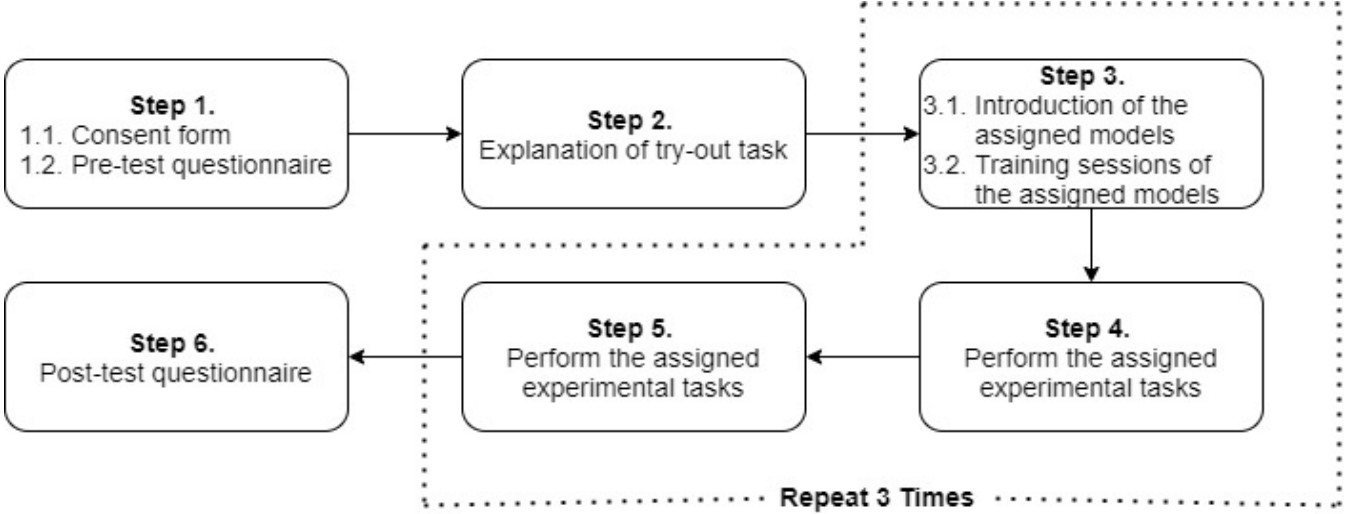

**Figure 4.** Experiment procedure.

### 4.4. Analysis Method

To answer the first research question, "Does the posting RS help users formulate questions in healthcare Q&A forums?", five measurements from the literature were used to evaluate outcomes: (1) input content length, (2) amount of medical terminology in input content [24], (3) presence of condition or self-description [25], (4) amount of recommended terminology adopted by user, and (5) total time to use to formulate post.

We used GEE [38] to analyze the data. Regardless of whether a variable data is continuous or nominal, GEE can estimate parameters. Even with missing data in a variable column, GEE can still calculate results from other columns containing data. GEE is suitable for repeatable experiments even if the input parameters are dependable or undependable and even if the population does not have a normal distribution. Lastly, the main effects and interaction terms of variables can be chosen under GEE manipulation.

To answer the second research question, "Is it easier for experts to understand questions supported by the posting RS?", we invited experts whose jobs were related to medical professions to rate the quality of posts composed by the participants. They noted that answering questions on a healthcare forum is similar to that of diagnosing patients in clinics. After patients describe the condition, professionals suggest possible solutions. The only risk is misunderstandings, as experts must judge illness given the posts alone, without face-to-face diagnosis.

Since an illness may present with different symptoms and complications in different people due to age, constitution, medical records, etc., it is difficult to draw conclusions when a replier sends ambiguous messages. To narrow down the range of possible solutions, it is necessary to obtain more details and transparent objectives (e.g., at least a query sentence and a self-description in a post). Therefore, when posting questions in the forum, the posting RS should assist users to adopt meaningful terminology and compose complete but concise posts. We requested professionals rate every post with one to five points (low to high quality) on three measurements: willingness, completeness, and clarity. Willingness evaluates whether the professionals were willing to answer c. Completeness and clarity concern the reason for their analysis. For example, informative contents (posts) were rated high in completeness, and contents (posts) with sufficient descriptions of what happened, as well as timing and location, were rated high in clarity.

## 5. Analysis of Results

The study was conducted with two posting RSs and one baseline model for three tasks. Each participant was requested to generate six posts in total. The system log was analyzed

to objectively investigate user behavior given the various RSs and task conditions, and participant posts were used to investigate whether recommendation support helps experts to better understand the posts and encourages them to answer the posted questions. This section is organized into log analysis and opinion analysis based on the research questions.

### 5.1. Log Analysis

This section focuses on the perspective of effectiveness from the post length, the number of medical-related features, and the existence of detailed descriptions among three models, the number of adopted recommended features between two experimental recommender models, and the perspective of efficiency between with RS and without RS. Table 1 shows a basic descriptive statistic of the three models. We applied a linear function with GEE to evaluate the association between post length and three within-subject variables: model, task, and operating order. There is shown to be no significant effect on models, but there is a main effect on tasks [$\chi(2)^2 = 11.758$, $p < 0.003$]. Investigating pairwise comparisons with the least significant difference (LSD) reveals that a significant difference in post length exists between allergy (mean = 63.02, S.E. = 5.381) and foodborne illness (mean = 45.72, S.E. = 4.096) ($p < 0.001$), and between flu (mean = 58.64, S.E. = 2.725) and foodborne illness ($p < 0.004$), suggesting that foodborne illness has a significantly shorter post length than both allergy and flu. However, no significant differences were observed between allergy and flu.

**Table 1.** Descriptive statistics of three main measurements among models. (Note. "A" denotes word embedding model, "B" denotes semantic model, and "C" denotes baseline model.).

| Model | Post Length | | | | Med-Related Word Count | | | | Existence of Descriptions | | |
|---|---|---|---|---|---|---|---|---|---|---|---|
| | Mean | S.E. | Min. | Max. | Mean | S.E. | Min. | Max. | True | False | TTL. |
| A | 59.87 | 26.937 | 18 | 154 | 3.87 | 1.602 | 0 | 9 | 35 | 19 | 54 |
| B | 62.81 | 34.575 | 19 | 189 | 4.56 | 2.661 | 0 | 12 | 33 | 21 | 54 |
| C | 60.93 | 33.389 | 15 | 149 | 3.89 | 2.724 | 0 | 16 | 35 | 19 | 54 |

Using GEE, a linear function was applied to evaluate the association between medical-related terminology and three within-subject variables: model, task, and operating order. This reveals a main effect for model [$\chi(2)^2 = 23.941$, $p < 0.000$], but no significant effects for task. A pairwise comparison with LSD reveals that participants composed posts using significantly more medical-related terminology ($p < 0.002$) with the semantic model (mean = 4.65, S.E. = 0.236) than with the word embedding model (mean = 3.21, S.E. = 0.560).

When a post includes more detailed context information, experts may better understand the user's questions and expectations. We asked the three curators to note whether posts contained descriptions about patient background and the timing of the illness outbreak. If the majority of the curators believed the post to be informative, it was labeled "T" (True); otherwise, it was labeled "F" (False). Using the GEE binary logistic function, we analyzed the association between the existence of descriptions and model, task, and operating order, revealing main effects of model [$\chi(2)^2 = 11.765$, $p < 0.003$] and task [$\chi(2)^2 = 25.799$, $p < 0.000$] on the existence of descriptions. In contrast to the baseline model (mean = 0.50, S.E. = 0.006), when using the word embedding model (mean = 0.48, S.E. = 0.006), participants were less likely to augment posts with descriptive information (OR = 0.910, $p < 0.002$). A pairwise model comparison demonstrated that participants were significantly less likely to add details when using the word embedding model than when using the semantic model ($p < 0.014$).

When comparing flu to the other two tasks, participants dealing with foodborne illness were more likely to add descriptions to their posts (OR = 1.107, $p < 0.004$). A pairwise task comparison indicated that (1) foodborne illness (mean = 0.51, S.E. = 0.006) and allergy (mean = 0.48, S.E. = 0.005) and (2) foodborne illness and flu (mean = 0.49, S.E. = 0.006) were significantly different. That is, participants were significantly more likely

to include descriptions in a post about foodborne illness than about allergy ($p < 0.000$) or flu ($p < 0.004$). There was no significant difference between allergy and flu in terms of the existence of descriptions.

We also evaluated which RS better supported users to generate posts by examining the adoption of two experimental models (word embedding and semantic) and the usage of medical-related terms. To gauge the quality of the embedding and semantic models, we first counted the number of adoptions during the asking process. From the viewpoint of total acceptance of the recommended terminology, the embedding model (43 times) yielded more than the semantic model (31 times). However, no significant effect was observed in model or task on the number of adoptions. Different from observing effectiveness of RSs, we also examined the amount of used time between RS (i.e., the embedding model or the semantic model) and the baseline model that is without recommendations with the GEE method. No significant effect was found, which demonstrates that users did not spend more time when using RSs compared to the baseline model.

In summary, in terms of effectiveness, applying an RS (i.e., the embedding model and semantic model) does affect asker posting behavior and encourages them to use medical-related terminology and include more description in posts. There was no significant relation between post length and whether askers used an RS. Participants were less likely to describe situations in detail when using word embedding than the semantic system. When analyzing tasks, the result shows that longer posts were used for foodborne illness than for allergy and flu. Participants included more details for foodborne illness scenarios than for allergy and flu. In terms of efficiency, applying RSs will not cost users more time to formulate their posts when they provide more details in their questions.

## 5.2. Opinion Analysis

We recruited two experts—one a pharmacist and the other a physician—to go through three lists of posts categorized by different tasks. Before asking the experts for their opinions, we interviewed them to determine how they judge their willingness to answer questions. Both experts indicated that complete and clear descriptions of conditions provide better information to help users. We use "willingness" to indicate their willingness to provide answers, and "completeness" and "clarity" as two factors that affect their willingness. The experts were asked to rate the three factors of posts on a five-point Likert scale (ranging from strongly disagree "1" to strongly agree "5"). The descriptive results of the three factors from the two experts are provided in Table 2.

**Table 2.** Descriptive statistics of three factors.

|  | Factor | Word Embedding | Semantic | Baseline |
|---|---|---|---|---|
| Expert 1 (pharmacist) | Willingness | $3.59 \pm 0.09$ | $3.52 \pm 0.09$ | $3.57 \pm 0.06$ |
|  | Completeness | $3.93 \pm 0.07$ | $3.98 \pm 0.08$ | $3.94 \pm 0.09$ |
|  | Clarity | $4.22 \pm 0.08$ | $4.17 \pm 0.08$ | $4.15 \pm 0.00$ |
| Expert 2 (physician) | Willingness | $3.83 \pm 0.09$ | $3.70 \pm 0.08$ | $3.96 \pm 0.09$ |
|  | Completeness | $2.98 \pm 0.11$ | $2.85 \pm 0.12$ | $2.93 \pm 0.07$ |
|  | Clarity | $3.20 \pm 0.10$ | $3.19 \pm 0.13$ | $3.20 \pm 0.14$ |

Inter-rater reliability with Cohen's Kappa [39] was adopted to evaluate the rating agreement of the two experts, yielding low Kappa values for willingness, completeness, and clarity [40], which could be attributable to their different backgrounds (pharmacist vs. physician), leading to different opinions in communicating with their patients [41]. Since there was no significant difference among the models for each expert, a linear function with GEE was applied to evaluate the association between willingness, completeness, and clarity and the existence of description separately on the expert judgment. The judgment of both pharmacist and physician showed that willingness (pharmacist: [$\chi(1)^2 = 22.194$, $p < 0.000$]; physician: [$\chi(1)^2 = 9.693$, $p < 0.002$]) and completeness (pharmacist: [$\chi(1)^2 = 62.246$, $p < 0.000$]; physician: [$\chi(1)^2 = 87.103$, $p < 0.001$]) are highly related to the existence of

description. The pairwise comparison of "False" and "True" label descriptions in the physician's willingness (False: mean = 3.61, S.E. = 0.42; True: mean = 3.78, S.E. = 0.40), the pharmacist's completeness (False: mean = 3.82, S.E. = 0.35; True: mean = 4.14, S.E. = 0.39), and the physician's completeness (False: mean = 2.32, S.E. = 0.73; True: mean=3.13, S.E. = 0.48) indicate that the "True" posts are more likely to get high points from experts. In terms of the effect of clarity on the existence of descriptions, a significant effect was found from the physician's judgment [$\chi(1)^2 = 36.817$, $p < 0.001$]. If askers did not include greater detail in posts, there was a 65.3% chance of getting less clarity points from the physician.

As we found that having a description contributes to higher points from experts, we directly investigated those posts with sufficient description between models to gain further insight. The judgment of both experts indicates that completeness is an important effect. The pairwise comparisons show that the semantic model is more likely to yield higher completeness points than word embedding and the baseline, whereas the word embedding model is more likely to get high completeness points from experts than baseline.

Clarity, the last measurement, was found to be significantly different between (1) word embedding and baseline ($p < 0.000$) and (2) semantic and baseline ($p < 0.002$). This suggests that using the posting RSs with sufficient post details is more likely to yield a high expert rating.

To examine the relationship between the quality of a user questions, the question's length (i.e., word count and med-related word count) and expert's opinions (including willingness, completeness and clarity) were investigated. As the recruited experts had diverse medical backgrounds and possessed non-identical perspectives, their opinions were therefore analyzed separately. The correlation results revealed that the experts' completeness and clarity were greatly affected by the word count and med-related word count (the results in Table 3 showed marginal differences in expert 1 and statistical differences in expert 2); however, experts' willingness was less likely to be influenced by the question length. In addition, the results indicated that the word count significantly impacted expert 2's opinions.

**Table 3.** Correlation results between word counts and expert opinions.

|  | Factor | Word Count | Med-Related Word Count |
|---|---|---|---|
| Expert 1 (pharmacist) | Willingness | Nonsignificant | Nonsignificant |
|  | Completeness | r = 0.136, p = 0.084 | Nonsignificant |
|  | Clarity | r = 0.145, p = 0.066 | Nonsignificant |
| Expert 2 (physician) | Willingness | Nonsignificant | Nonsignificant |
|  | Completeness | r = 0.468, p < 0.001 | r = 0.266, p = 0.001 |
|  | Clarity | r = 0.192, p = 0.014 | r = 0.248, p = 0.001 |

## 6. Discussion

It is easy to find online Q&A forums with mechanisms to support finding existing relevant questions, but it is hard to find supportive systems that focus on post composition during the query process. This study demonstrates that the proposed posting RSs are more effective and efficient than the baseline (with no RS support).

The amount of medical-related terminology has a significant effect on models, showing that using an RS yields more medical-related terminology compared to when an RS is not used. However, the sematic model has a stronger influence than the embedding model, whereas the word embedding model usually yields more relevant topics based on common wordings than the semantic dictionary-based corpus. The semantic corpus, constructed by manipulating WordNet, performs well particularly when askers are able to query more professionally. The weaker performance of the embedding model might be due to the small training dataset, leading to imprecise or ambiguous recommendations. To improve the usefulness of the word embedding model, the future work must collect larger amounts of in-domain data and then re-train the model.

Detail in a post is an important element for experts to evaluate posts because they cannot diagnose a person via back-and-forth interaction: A single question is usually not enough for experts to solve the problem. Our data reveal the main effects between having descriptions on models and tasks. A deeper investigation indicates the embedding model is less likely to result in more details in a post, in contrast to the baseline. This indicates that people are still used to a posting procedure without interference. In addition, as it is merely a simulated scenario, most participants lacked a strong motivation to find a solution. They feel more comfortable writing posts in a stress-free situation without interruptions. In addition, as allergy and flu are common experiences, participants may assume that most readers are familiar with them and thus omit details when describing the malady. In contrast, when generating posts about foodborne illness, which is less familiar, participants provided more details when describing the conditions. Post length was not found to differ significantly between models but it did between tasks, which indicates that different illnesses do affect post length. The interviews revealed that most participants are not familiar with foodborne illness; this unfamiliarity caused participants to compose posts that were shorter than those for allergy and flu.

Suggestions from the embedding and semantic models were adopted 43 and 31 times, respectively. However, each model had 54 posts and adoption was unevenly distributed in each post: For many posts, none of the recommended terminology was selected. This indicates more resources would be needed in the future work to build a robust word embedding model. If a recommendation looks strange, even though the average score for "want to use this kind of topic RS someday" was 3.81/5.00 in the post-questionnaire, poor user experiences dictate that it would be difficult to attract attention.

To explain the connection between a post RS and higher scores from experts, we further conducted a pairwise evaluation between the interaction of models with descriptions labeled "True" and three measurements. According to the result of the first expert (the pharmacist), completeness is higher when using a posting RS with detailed descriptions. Completeness and clarity of the second expert (physician) are increased if an asker uses the RS and provides more details in a post. Although results vary between experts, we conclude it is possible to elicit a response from experts after using an RS and adding details. In addition, we found that willingness is not significantly affected by a post RS that adds details, because the professional ethic of medical experts is to answer patient questions; thus, they seldom refuse to answer such requests. Therefore, willingness may not be a good measurement.

As both physicians and pharmacists are highly specialized and regulated professions, through the rigorous medical training, we assume individual differences in attitude would inject little influence of the collected expert opinions. Therefore, in terms of expert opinions, since professionals from different disciplines have different norms in communicating with their patients, it is difficult to find common ground between physicians and pharmacists [41]. For physicians, the priority is to thoroughly understand the situation and any information that relates to the patient's symptoms [41,42], whereas as pharmacists tend to focus on medicinal instructions and materials; it is more important for them to gather all of the critical information than to understand the situation as a whole [41]. Despite the marked difference between the two experts' evaluations, both pharmacist and physician consider willingness and completeness to depend greatly on the existence of sufficient detail in the problem description. Posts labeled "False" are less likely to earn points from the expert. The physician's judgment also demonstrates that clarity is an important factor as well. According to the interviews with experts, getting a good score from the physician means the post is easily understood by experts. Easily comprehensible posts are more likely to be solved. Although some interesting results are observed, however, due to the small number of the experts used in this study, future work should address this issue and recruit more medical specialists to further validate our findings as well as exclude any potential issues that may arise from the sample size limitation.

### 7. Conclusions

In this work, we present a post RS that suggests relevant and useful ideas and terminology to support users who are composing posts to ask questions. Effectiveness and efficiency are evaluated in terms of the usability of the proposed post RSs (RQ1). Combining the result with RQ1, we evaluate the feasibility of the resultant posts to see if experts assign them higher scores (RQ2).

This research reveals that current Q&A forum RSs have reached a plateau because they only recommend relevant questions based on the words in the query and then send query requests to those who might be able to help the askers. These supportive methods may be infeasible when posts are difficult for the system to classify and users may decline to bother people who are reluctant to answer. In addition, most Q&A online forums do nothing about post actions in the asking process. Also, the existence of unanswered posts underlines the necessity of optimizing the posting process. After this user study, we found it is possible to change user posting behaviors by participating more in the asking process via a posting RS. Askers are also willing to be supported by the RS feature when formulating questions in unfamiliar domains. Whether the recommended terminology can be adopted directly or is relevant enough to modify posts conceptually, our RS suggests concrete and possible ideas to askers, which constitutes a new type of manipulation in the Q&A domain. We therefore anticipate that the posting RS will support users to better formulate posts and find solutions in a more efficient manner.

The proposed posting RS is also applicable to domains other than healthcare. Take e-commerce for example: when people are purchasing products that they are not familiar with, it is common for them to ask for details before and after the purchase. If there were a system that would help users compose better questions, the resultant posts would better match the FAQs. If solutions are still not found in the FAQs, websites present previous posts from other askers. An advanced posting RS could attempt to resolve questions before posting to the forum. The unanswered rate would decrease and the likelihood of getting a solution would increase. Any industry that fields many queries is suitable for more participation in the user's asking process.

For the future work, the number of participants should be increased, the illness selection should be reconsidered, and the data resources to make a RS should be expanded. The recommendation presentation should be made more user-friendly. Second, some participants felt the selected tasks to be so general that they did not need an RS to complete the post, whereas others considered the selected tasks too difficult to compose a post about, suggesting feedback varied widely among participants. In the future, a study with various tasks and more participants might be able to bring us more insights for designing the posting recommender systems. Also, the quality assessment of our posting RS is important. Collecting more data from healthcare forums is the most direct way to improve the performance of posting RSs. However, what kind of data resources should be selected to build the posting RS? If the quality of the input (existing posts on online forums) is low, there would be little chance of producing a high-quality RS. Therefore, training models on high quality posts is one way to enhance the usefulness of the RS.

Regardless of whether the RS data sources support high quality revisions, the quality of posting RSs should be evaluated in advance. One potential approach is to take the first sentence of good WebMD questions to see whether the proposed RS can suggest sufficient terminology to formulate the subsequent sentences. Sufficient terminology could be identified by mapping the recommendations to the rest of the sentences of good questions. Then we could observe if the relevant terminology suggested matches the terminology used in the subsequent sentences of every post. Further study with eye-tracking augmentation could be useful to learn more about interactions between the process of decision-making and types of posting RSs.

In addition, while posting recommendations can help to compose posts in a more detailed way to attract experts to answer, the more detailed content provided the more sensitive data releases online. This is always a dilemma between efficiency and privacy.

Practitioners might need to pay attention to the forum policy when providing a posting recommender system.

**Author Contributions:** Y.-L.L.: conceptualization, methodology, writing—original draft. S.-Y.C.: methodology, writing—review & editing. Y.-J.C.: software, validation, formal analysis, investigation, data curation. All authors have read and agreed to the published version of the manuscript.

**Funding:** This research was supported by the Ministry of Science and Technology, Taiwan, under Grant MOST 107-2410-H-004-098-MY3 and MOST 109-2410-H-004-067-MY2.

**Institutional Review Board Statement:** The study was conducted according to the guidelines of the Declaration of Helsinki, and approved by the Research Ethics Committee of National Chengchi University (protocol code: NCCU-REC-201709-I036; date of approval: 16 August 2019).

**Informed Consent Statement:** Informed consent was obtained from all subjects involved in the study.

**Data Availability Statement:** The data presented in this study are available on request from the corresponding author. The data are not publicly available due to privacy concerns.

**Acknowledgments:** The authors thank Tsung-Hua Shen, a research assistant in the College of Pharmacy at Taipei Medical University, and Po-Yu Liao, a Physician at Liao ENT Clinic, for their valuable assistance in reviewing and categorizing the participants' queries in the opinion analysis phase.

**Conflicts of Interest:** The authors declare no conflict of interest.

### Appendix A  Supportive Paragraphs for Participants

| | |
|---|---|
| *[Task 1]* | **Foodborne Illness**<br><br>Lisa just got a call from her aunt who is a nurse in a nearby hospital. Many students were sent to the emergency room this afternoon because of a foodborne illness. It is said that the food vendor failed to check the expiration date for their meat and sent it to several chain restaurants. Unfortunately, Lisa's favorite restaurant gets meat from this vendor and she just went there for brunch. Just to be safe, Lisa looked up information on food contamination on Google. The following paragraph is what she found. Please write down the questions you would ask on the forum if you found yourself in a similar situation. You may post either as yourself or Lisa. |

Food poisoning symptoms vary with the source of contamination. Most types of food poisoning cause nausea, vomiting, watery or bloody diarrhea, abdominal pain, and cramps and fever. Signs and symptoms may start within hours after eating the contaminated food, or they may begin days or even weeks later. Sickness caused by food poisoning generally lasts from a few hours to several days. Sometimes, there are serious complications. Whether you become ill after eating contaminated food depends on the organism, the amount of exposure, your age and your health. High-risk groups include older adults, pregnant women, infants and young children, and people with chronic disease, who are highly affected by their immune system or changes in metabolism and circulation. Food poisoning is especially serious and potentially life-threatening for them. At home people can stay safe by taking preventions such as separating raw foods from ready-to-eat foods, washing hands before eating, and defrosting foods safely.

**Foodborne Illness**

*[Task 2]*

You are doing a term project related on foodborne illnesses. The professor has asked you to organize questions for a class discussion and to post them on the Online Discussion Board before next week's class. You may share your opinions in the post. You may also propose questions, for instance, concepts you didn't understand after reading the supportive paragraph, or alternatively, guess what questions corresponds to the concept.

Food poisoning syndrome results from the ingestion of water and a wide variety of food contaminated with pathogenic organisms (bacteria, viruses, parasites, and fungi), their toxins and chemicals. Food poisoning must be suspected when an acute illness with gastrointestinal or neurological manifestations affects two or more persons or animals who have shared a meal during the previous 72 h. The term generally used encompasses both food-related infection and food-related intoxication. Some microbiologists consider microbial food poisoning to be different from foodborne infections. In microbial food poisoning, the microbes multiply readily in the food prior to consumption, whereas in foodborne infection, food is merely the vector for microbes that do not grow on their transient substrate. Other consider food poisoning as intoxication of food by chemicals or toxins from bacteria or fungi.

Foodborne illness (FBI), often called food poisoning, is caused by pathogens or certain chemicals present in ingested food bacteria, viruses, molds, and worms. Protozoa causing diseases are all pathogens, although there are also harmless and beneficial bacteria that are used to make yogurt and cheese. Some chemicals that cause foodborne illness are natural components of food, whereas others may be accidentally added during production and processing, either through carelessness or pollution. The two most common types of food borne illness are intoxication and infection. Intoxication occurs when toxins produced by the pathogens cause food poisoning, whereas infection is caused by the ingestion of food containing pathogens.

*[Reference]*

https://www.omicsonline.org/open-access/a-review-on-major-food-borne-bacterial-illnesses-2329-891X-1000176.pdf

https://www.mayoclinic.org/diseases-conditions/food-poisoning/symptoms-causes/syc-20356230

Allergy

*[Task 1]*

Steven has a nasal allergy. When the weather changes, his illness gets worse. Yesterday, his sneezing was so bad he went through a whole tissue box in 20 minutes! Interestingly, Steven recently discovered his 5-year-old son is allergic to seafood, especially crab and shrimp. If the food is not fresh enough, his son gets an itchy rash all over his body - symptoms totally different from his own. Although they have taken medicine for allergies, Steven wonders if they need to see a doctor. The paragraph below is an overview he found on Google. Please think of questions you would ask on the forum if you found yourself in a similar situation. You may post either as yourself or Steven.

Some people suffer with seasonal allergies for years before learning about effective treatments. If allergy symptoms are not treated early, they can actually worsen over time. Here are five symptoms you should not ignore: runny or stuffy nose, sinus pressure, sneezing, itchy eyes, and postnasal drip. You may avoid your allergy triggers or ask doctors about other ways to get relief. Food allergies are an immune system reaction that occurs soon after eating a certain food. It is easy to confuse a food allergy with a much

more common reaction known as food intolerance. While bothersome, food intolerance is a less serious condition that does not involve the immune system. Itching in the mouth, swelling of the lips, face, or other parts of the body, etc., are common signs of the food allergies. People who have similar symptoms should keep away from food triggers, for example, shellfish, peanuts, and fish.

---

**[Task 2]**

Allergy

You are doing a term project related to world allergy proportions. The professor has asked you to organize questions for a class discussion and to post them on the Online Discussion Board before next week's class. You may share your opinions in the post. You may also propose questions, for instance, concepts you didn't understand after reading the supportive paragraph, or alternatively, guess what questions corresponds to the concept.

---

Allergies involve almost every organ of the body in variable combinations with a broad spectrum of possible symptoms; thus, their manifestations cover a wide range of phenotypes. Studies in Europe have shown that up to 30% of the population suffer from allergic rhinoconjunctivitis, whereas up to 20% suffer from asthma and 15% from allergic skin conditions. These numbers match those reported for other parts of the world, such as the USA and Australia. Food allergies are becoming more frequent and severe; occupational allergies, drug allergies, and allergies to insect stings (occasionally fatal) further aggravate the burden of the allergy epidemic. Despite the popular belief that allergies are mild conditions, a considerable and increasing proportion of patients (15–20%) have severe, debilitating disease and are under constant fear of death from a possible asthma attack or anaphylactic shock. Within the EU, there are nevertheless wide geographical variations in the incidence of allergies with a south to north and east to west gradient. An alarming observation is that most allergic conditions start in childhood and peak during highly productive years of individuals, with allergic rhinitis affecting up to 45% of 20 to 40-year-old Europeans. The numbers may even be an underestimation, as many patients do not report their symptoms or are not properly diagnosed. Indeed, it is estimated that approximately 45% of patients have never received a diagnosis. Notwithstanding evidence suggesting a plateau in some areas, the European Academy of Allergy and Clinical Immunology (EAACI) warns that in less than 15 years more than half of the European population will suffer from some type of allergy!

*[Reference]*

https://www.ncbi.nlm.nih.gov/pmc/articles/PMC3539924/

https://www.webmd.com/allergies/features/allergy-symptoms#2

https://www.mayoclinic.org/diseases-conditions/food-allergy/symptoms-causes/syc-20355095

---

**[Task 1]**

**Flu**

In Tommy's school, one in four students has come down with the flu, so the junior high school committee has claimed it is necessary to close the school for disinfection. Tommy's mom is concerned about the symptoms of this flu because she forgot to have Tommy get the vaccination this year. Please think of questions you would ask on the forum if you were in a similar situation. You may post either as yourself or one of Tommy's parents. The following information seems useful to Tommy's mom. Feel free to refer to it if you need ideas about what to say.

---

I. Seasonal influenza (or "flu") is most often caused by type A or B influenza viruses. Symptoms include a sudden onset of fever, cough, headache, muscle and joint pain, sore throat, and a runny nose. The cough can be severe and can last 2 or more weeks. Most

people recover from fever and other symptoms within a week without requiring medical attention. However, influenza can cause severe illness or death in high-risk groups.

II. Someone with the flu may have a high fever, for example, their temperature may be around 104 °F (40 °C). People with the flu often feel achy and extra tired. They may lose their appetites. The fever and aches usually disappear within a few days, but the sore throat, cough, stuffy nose, and tiredness may continue for a week or more. The flu also can cause vomiting, belly pain, and diarrhea. Most people who get the flu get better on their own after the virus runs its course. However, call your doctor if you have the flu and any of these things happen: (a) you are getting worse instead of better; (b) you have trouble breathing or develop other complications, such as a sinus infection; or (c) you have a medical condition (for example, diabetes, heart problems, asthma, or other lung problems). Most teens can take acetaminophen or ibuprofen to help with fever and aches.

---

*[Task 2]*

**Flu**

You are doing a term project on the flu. The professor has asked you to organize questions for a class discussion and to post them on the Online Discussion Board before next week's class. You may share your opinions in the post. You may also propose questions, for instance, concepts you didn't understand after reading the supportive paragraph, or alternatively, guess what questions will be provided by the human society to find information like the following paragraphs.

---

What scientists dream of is a vaccine that can protect against any flu strain for years or even a lifetime. This so-called universal flu vaccine is still a long way off, if it is even possible. However, many labs are dusting off past projects on broad flu vaccines, spurred by new funding and fears that H5N1, the deadly avian influenza that has swept across half the world, could acquire the ability to be transmitted from human to human. Until now, "flu has never been before high enough on the radar screen" for companies in particular to follow through with a strong push for a universal vaccine, says Gary Nabel, director of the Vaccine Research Center at the U.S. National Institute of Allergy and Infectious Diseases (NIAID) in Bethesda, Maryland.

Doing so, however, means coming up with an alternative way to stimulate immunity to the virus. The tried-and-true technique for seasonal flu uses a killed virus vaccine that works mainly by triggering antibodies to hemagglutinin (HA), the glycoprotein on the virus's surface that it uses to bind to human cells. Hemagglutinin and neuraminidase (NA), another surface glycoprotein that helps newly made viruses exit cells, give strains their names (H5N1, for example). The sequences of HA and NA mutate easily, which is why each season's flu strain—although it may be the same in subtype, such as H3N2—"drifts" slightly from the previous year's, and the annual vaccine must be tailor-made.

To make a universal vaccine for influenza A, which includes the main seasonal flu strains and bird flu, as well as past pandemic strains, some scientists are hoping to use "conserved" flu proteins that do not mutate much year to year. (Influenza B, the other type, occurs only in humans and causes milder symptoms.) Some of the conserved protein vaccines in the works stimulate the production of antibodies as do conventional flu vaccines, whereas others rouse certain immune system cells to battle the virus.

*[Reference]*

http://science.sciencemag.org/content/312/5772/380
http://www.who.int/features/qa/seasonal-influenza/en/
https://kidshealth.org/en/teens/flu.html

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
