# Peer review of "Posting Recommendations in Healthcare Q&A Forums"

_electronics, doi:10.3390/electronics10030278_

Round 1

Reviewer 1 Report

Authors should include scientific explanation of different machine learning models used i.e. Word2Vec.  Readers of journal "electronics" are (generally) inclined towards scientific novelty in the domain of computers which is lacking in the article.

Reviewer 2 Report

The paper presents important and interesting in principle research. But what concerns current presentation I want to make several remarks. 

Any recommendation system needs to be evaluated according two main aspects - first, quality of the user's question (clarity, completeness, etc.) and second, efficiency of the user's question (e.g. shorter text likely would be better if it is possible to understand it properly). It is important that the evaluation will be performed in this order. It remains unclear if fewer words used in questions about e.g. foodborne illnesses had impact on the clarity of user's questions or not. If we don't have evaluation of the clarity of the user's questions then the data in Table 1 has little value (as in whole chapter 5.1). So in my opinion first authors needs to evaluate the quality of the questions in the different groups of illnesses and only then go to the evaluation of the efficiency. 

Next remark is related with the method used in the opinion analysis (chapter 5.2). Authors used two experts - pharmacist and physician - to express their views (clarity, completeness) about the questions. When you have single representative of each profession it is very hard to find which aspects are dependent on professional background and which are caused by personal  attitudes of the people. Here I would like to suggest to try apply differential analysis approach (very widely used in humanities and social sciences): to try to invite two representatives of each profession. Such approach allows to reduce significantly impact of personal attitudes to the inevitably subjective questions and to clarify what is caused by the professional background rather than personal views. If this is impossible I suggest to avoid carefully any statements about relation of the obtained results to the professional background of expert. 

If these problems will be taken into the account the conclusions made in the paper will sound better grounded. Some of them may need reformulation. Also I would like to recommend to avoid using such conclusions as this "RS system may help non-native English speakers to formulate" better question content. The paper not addressed at all the issue of the question formulation of native and non-native English speakers and the impact of this RS system to non-native English speakers is unclear (or the results aren't provided). 

In the chapter about related work authors can avoid some quite obvious citations about the general usefulness of RS systems but to provide more details about particular methods used in similar systems, especially which methods proved to be more efficient which not so efficient (not necessary in the healthcare oriented RS). 

Despite those remarks I am sure that authors are able to do necessary changes and then the paper can be published

Reviewer 3 Report

I really liked idea of using RS for improving quality of postings to Q&A forums. The design of research and validation methods are adequate. 

I would recommend to consider evaluation of personalized RS in further research as authors noticed that participants considered difficulty of given tasks to have different levels.

I would also recommend to include more up to date references. 

Reviewer 4 Report

The paper tackles issues that are extremely relevant in the context of the pandemics. The authors present a documented and well-structured research on posting recommendation systems that aims to streamline communication between askers and experts. A brief analysis of Q&A forums used to ask and answer technical questions might provide a useful reference point for comparison in the context of this research. Several key challenges are identified and discussed. Results are presented clearly. Since the RSs recommend ideas, it would be useful if more information would be provided on how the RSs are able to learn and, for example, how is the validity of the data sources verified in order to avoid misinformation and misleading. Also, since healthcare is a special domain that is associated with specific responsibility and critical risks, how are recommendations mitigated to deal, for example, with the safety of the asker, avoiding revealing sensitive data, the effects of the hypochondriasis syndrome?

Round 2

Reviewer 1 Report

Authors have significantly improved content of the article by addressing raised queries.

Reviewer 2 Report

This version of paper is partially improved comparing with the previous version. In my opinion is still possible to improve the presentation of results but let to leave this for the authors: anyway presentation is good enough to be published.
